# Few-Shot Learning for Industrial Defect Detection on Novel Scanning Electron Microscopy Datasets

Anonymous Full Paper
Submission 34

## Abstract

Industrial Defect Detection (IDD) involves identifying defects in different products through the analysis of manufacturing data. Over recent years convolutional neural networks (CNN) have become the preferred method to reliably solve this task, though a lack of labeled data has been a key challenge for supervised methods that rely on CNNs. Few-Shot Learning (FSL) offers a promising solution by enabling models to learn tasks from only a small number of labeled examples. However, it shifts the need for large labeled datasets to the pre-training stage, raising questions about how well these models generalize to new domains, such as different imaging modalities. Therefore, this study evaluates state-of-the-art FSL methods, trained on public optical datasets, for their effectiveness in IDD when tested on scanning electron microscopy (SEM) images. To facilitate benchmarking, this article also introduces three distinct SEM datasets for defect detection purposes. Through this assessment the study is able to identify strengths, challenges, and potential areas of improvement to motivate further research.

**Keywords**: Industrial defect detection, convolutional neural networks, few-shot learning, scanning electron microscopy images.

## 1 Introduction

Recent developments in the field of computer vision have significantly accelerated the creation and training of automated systems to inspect industrial products at several stages of manufacturing. Inspection and analysis of such data not only enable quality control on the production line but also the detection of key areas to potentially improve existing manufacturing processes [1, 2]. Industrial defect detection (IDD) presents a task to detect anomalies in industrial manufacturing data that manifest themselves in modalities such as optical inspection (grayscale or color), X-ray imaging, or electron microscopy images of components, parts, assembly, and tiny semiconductor structures. In this domain, the current state of the art presents itself in supervised methods that rely on Convolutional Neural Networks (CNNs) and large amounts of task-specific labeled data [3, 4]. Since reviewing and manually labeling images for every task is necessary but time-consuming and labor-intensive task in the case for supervised methods, models that can quickly learn a new task from just a few labeled examples are particularly valuable. This motivates the use of few-shot learning (FSL) for IDD, where methods aim to adapt to previously unseen tasks using only a small number of annotated images. As FSL methods have been shown to be competitive with standard supervised models in a classification setting [5] and semantic segmentation for IDD [2], this paper uses FSL methods in a classification setting to detect industrial defects. However, FSL does not eliminate the need for large labeled datasets; it merely shifts the problem from training a task-specific model to training a more general model capable of handling a wide range of tasks. In practice, this often involves relying on publicly available, large-scale optical image datasets for pre-training. These datasets must capture the underlying characteristics of the target tasks to be effective, yet while there is an abundance of data for some modalities, such as natural images, there is very limited publicly available data for others, such as defect detection in SEM images. In this work, we aim to develop an understanding of how well FSL methods can be adapted to defect detection from an industrial perspective by benchmarking these methods on real-world scanning electron microscope (SEM) images provided to by [COMPANY NAME OBSCURED FOR BLIND REVIEW]. In tandem, this would also provide an outlook on how well models are able to generalize when trained on optical images but benchmarked on SEM images. The contributions of this paper can be summarized as follows:

- Publication of three scanning electron microscopy datasets for the purposes of industrial defect detection at the courtesy of [COMPANY NAME OBSCURED FOR BLIND REVIEW].

- Proposal of a new evaluation framework to evaluate few-shot learning methods in the context of industrial applications for defect detection.

- Benchmarking insights on how few-shot learning methods work when trained on publicly available data and applied on real-life industrial data from a different imaging modality.

## 2 Methods

### 2.1 Few-Shot Learning Problem Formalization

The main aim of few-shot learning (FSL) is to extract transferable knowledge from an auxiliary data set $A$ to adapt to a new unseen task T with a small amount of labeled samples. For example, given a dataset $D$, it's split into $D_{train}$, $D_{val}$, and $D_{test}$ to train and evaluate FSL methods. Each split is class disjointed; meaning, that it is possible to define sets of classes $C_{train}$, $C_{val}$, and $C_{test}$ exclusively belong to their own split. $D_{train}$ is utilized as $A$, which is a class-rich split and is assumed to be within the same domain as $D_{val}$ and $D_{test}$. A task $T$ is defined by two sets: a labeled support set and an unlabeled query set. The support set $S$ contains $N$ classes with $K$ samples per class, which is why few-shot learning tasks are commonly described as N-way-K-shot in classification settings. To evaluate a model's ability to learn from limited samples, such tasks $T$ are randomly sampled from $D_{val}$ and $D_{test}$. A query set $Q$ is comprised of $K$ samples per class from the same label space as $S$ and is utilized to evaluate the performance of a given method for a specific task.

### 2.2 Few-Shot Learning Methods Evaluated in This Study

This paper evaluates a total of six few-shot learning methods (mentioned in Table 1.) which span two major categories: non-episodic and episodic methods. Non-episodic methods are similar to transfer learning methods in which a model is pre-trained on the auxiliary set and then fine-tuned during inference on the given support set [6]. Episodic methods are further divided into two subtypes: meta-learning and metric learning. Meta-learning aims to learn with small gradient steps to quickly adapt to an unseen new task. Metric learning methods adapt to a new task by comparing the distances between images from the support and query set images [6]. The methods evaluated in this paper are as follows: Baseline is a non-episodic transfer learning approach; the model is pre-trained on auxiliary data and then fine-tuned on target tasks [7]. SKD (Gen-0) is another non-episodic method that employs self-supervision with data augmentation to learn a broad classification manifold [8]. Among episodic meta-learning methods, MAML trains models to rapidly adapt to new tasks using only a few gradient steps through meta-learning [9]. BOIL also uses episodic meta-learning but differs by updating only the feature extractor during the inner-loop updates [10]. In episodic metric learning, ProtoNet classifies by computing distances to prototype representations of each class [11], while RENet enhances metric learning with self-correlation and correlational attention to capture rich semantic relationships between query

and support sets.

### 2.3 Implementation Details

The methods discussed above are implemented by Li et al. [6] in an open-source project called LibFewShot, which aims to facilitate research within the few-shot learning (FSL) paradigm. Built in PyTorch [12], LibFewShot provides re-implementations of 18 state-of-the-art FSL methods with multiple backbones and a flexible yet reproducible framework to perform comparative experiments. Following common practice [6, 11, 13], all images in this study (training and evaluation) are resized to 84 x 84 pixels and normalized as preprocessing, and the LibFewShot training procedure is followed where the performance on the validation set is monitored to save the best performing model. Although methods commonly use Conv64F, ResNet-12, and ResNet-18 as backbones [6, 10, 14], for this study, the simplest backbone (Conv64F) is selected based on time feasibility and computational resources available. The architecture of Conv64F includes four identical blocks where each block is composed of a convolutional layer with 64 3x3 filters, a batch normalization layer, a ReLU / LeakyReLU layer and an optional max-pooling layer [6, 11]. Each method has a preferred optimizer and hyperparameters, LibFewShot provides suggestive configurations, which are followed for this study. Following common practice [9, 11], for each method mentioned, models are trained in the 1-shot and 5-shot scenarios. Then the models are evaluated in 1-shot, 5-shot, 10-shot, and 20-shot scenarios.

**Table 1.** Methods evaluated in this study.

| Method | Category | Subtype | Reference |
|---|---|---|---|
| Baseline | Non-Episodic | - | [7] |
| SKD (Gen-0) | Non-Episodic | - | [8] |
| MAML | Episodic | Meta-Learning | [9] |
| BOIL | Episodic | Meta-Learning | [10] |
| ProtoNet | Episodic | Metric Learning | [11] |
| RENet | Episodic | Metric Learning | [13] |

### 2.4 Metric and Evaluation Framework

#### 2.4.1 Metric

As a key challenge in defect detection tasks is the high imbalance between classes, where defect samples tend to be rarer than non-defect ones, accuracy as a metric can be misleading due to the over-representation of the majority class in the test dataset [15]. To illustrate this issue, consider a test dataset with 990 non-defect images and 10 images with defects. If a given classifier predicts all 1000 images of the test set to be non-defect, it would automatically achieve an accuracy of 99%, which fails to capture the failure of the classifier to predict the minority class entirely. To account for a classifier's ability to predict the majority and minority classes,

this study adopts balanced accuracy as the reported metric, which can be described as the average recall for all classes from a dataset (eq.1). This ensures that each class is represented equally regardless of its prevalence in the dataset [15]. For a multi-class classification problem with $N$ distinct classes, the balanced accuracy is defined as:

$$\text{Balanced Accuracy} = \frac{1}{N} \sum_{i=1}^{N} \frac{TP_i}{TP_i + FN_i}, \quad (1)$$

where $TP_i$ and $FN_i$ are the numbers of true positive and false negative predictions, respectively, for the $i$th label.

### 2.4.2   Evaluation Framework

LibFewShot provides a standardized framework for evaluating FSL methods. It assesses models by sampling 3,000 balanced tasks from a data split and reporting the mean accuracy with a 95% confidence interval [6]. This standardized process enables consistent benchmarking in academic research; however, because classes are randomly sampled, tasks may include unrelated classes. As this study is interested in measuring the performance of models for industrial applications on specific tasks, the evaluation strategy was updated accordingly in the following way: Instead of sampling multiple tasks from a dataset, the entire evaluation dataset is turned into one task, where a support set is defined of K-shots from N-classes from the dataset (illustrated in Fig. 1), and the remainder of it is considered a query set. This enables us to estimate the approximate performance of a model for a given dataset when using a few-shot learning method. To capture the true range of results over multiple support sets, the updated evaluation process described in this paper is repeated 50 times with randomized support sets, and the mean of balanced accuracy and the 95% confidence interval are reported. The evaluation process is conducted on all test datasets sets individually to record accuracy per dataset. Although these fifty evaluations conducted are significantly lower than the standard of 3000 evaluations in LibFewShot, query sets are commonly limited to 15 images per class in a task during regular evaluation [11], while query sets in this study are significantly larger. This allows the study to leverage more data to provide representative results while addressing computational constraints.

## 3   Experiments & Results

### 3.1   Datasets

For conducting experiments, the Industrial 5i dataset [1] is utilized as the auxiliary and validation set containing 80% and 20% of classes, respectively. The Industrial 5i dataset (Fig. 6), introduced by

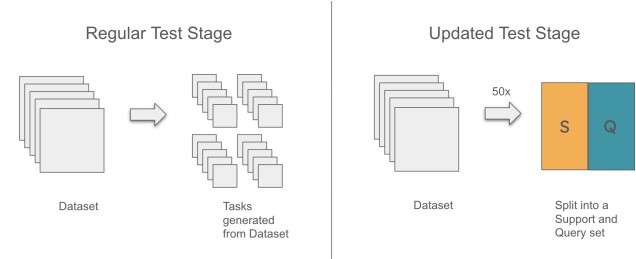

**Figure 1.** Updated evaluation framework for this study illustrated.

[1] was originally proposed for a K-shot image segmentation task and comprises common industrial datasets such as MVTEC-AD [16], KolektorSDD [3], etc. Each product has a positive class (non-defect) and negative class (defect) and a corresponding binary mask as a label for semantic segmentation. The data set is converted into a classification task to align with the objective of the project and then used for training and validation. With 20 products and 2 classes per product, the dataset is recreated with 40 total classes and to prevent data leakage between classes of the same product, the train, and validation splits are randomly allocated distinct products instead of classes. The product-wise split is followed to ensure a model learns to distinguish between good and defective classes within the same dataset.

### 3.1.1   Scanning Electron Microscope Datasets

The three test datasets are produced by and published at the courtesy of [COMPANY NAME OBSCURED FOR BLIND REVIEW]. Each dataset represents a unique pattern created by electron-beam lithography, with bright areas being the silicon wafer trench floor and unexposed PMMA (polymethyl methacrylate) resist being the dark area. The data sets are captured with an SEM detector biased to collect as many electrons as possible, resulting in additional brightness around any shape edge present in the data set. These data sets differ from each other in a key characteristic. 'Trench' was created specifically to test models with artificially added defects that are positionally consistent from image to image. 'Lines & Spaces' and 'Josephson Junction' were created for meteorological studies and the labels are added through manual annotation of naturally occurring defects. Figures 3, 4, and 5 show samples of each class present within the data sets, and the class distributions of the evaluation datasets are illustrated in Fig. 2. The descriptive statistics for each split of the dataset are mentioned in Table 2.

### 3.2   Results

For each method mentioned, a model is trained in the 5-way-1-shot and 5-way-5-shot scenarios. They

**Table 2.** Descriptive Statistics of Data Splits.

| Split | # Unique Classes | # Images |
|---|---|---|
| Train/Auxiliary | 30 | 16,070 |
| Validation | 10 | 1,111 |
| Test | Trench: 6, JJ: 3, L&S: 2 | Trench: 1,200, JJ: 1,000, L&S: 1,000 |

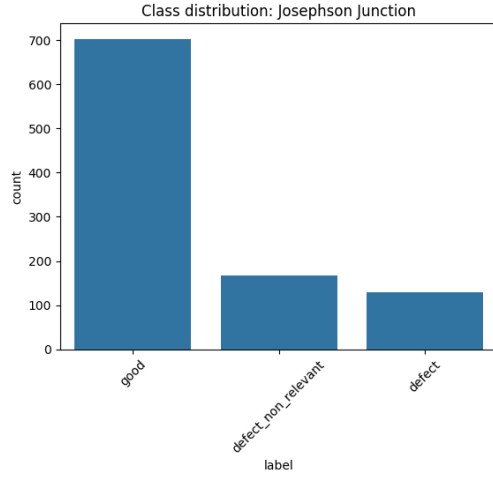

Josephson Junction

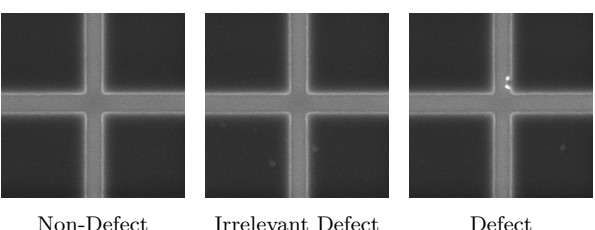

Non-Defect   Irrelevant Defect   Defect

**Figure 3.** Classes from the Josephson Junction dataset.

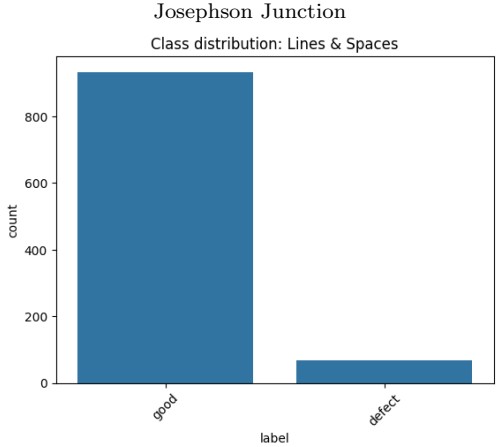

Lines & Spaces

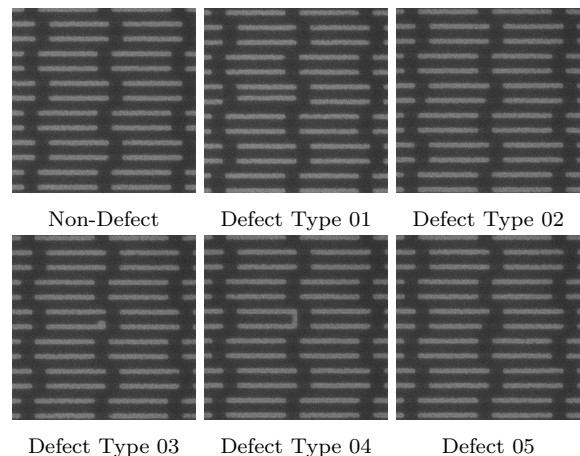

Non-Defect   Defect Type 01   Defect Type 02

Defect Type 03   Defect Type 04   Defect 05

**Figure 4.** Classes from the Trench dataset.

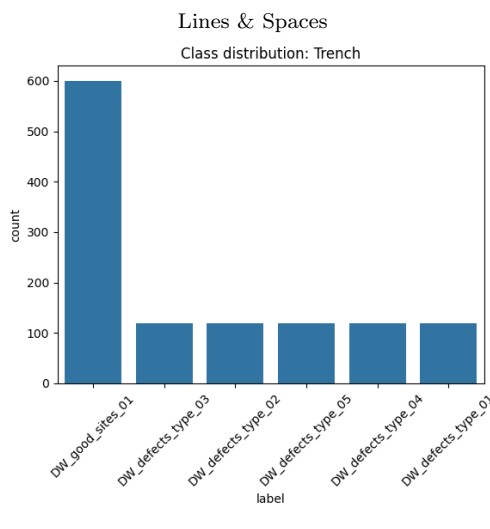

Trench

**Figure 2.** Class distributions of the evaluation datasets.

 

Non-Defect          Defect

**Figure 5.** Classes from the Lines & Spaces dataset.

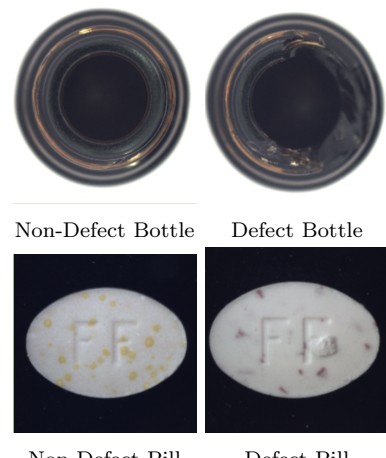

Non-Defect Bottle  Defect Bottle

Non-Defect Pill  Defect Pill

**Figure 6.** Examples of products from Industrial 5i.

are then evaluated in the 1-shot, 5-shot, 10-shot, and 20-shot scenarios with each dataset. The reported metrics as shown in Tables 3, 4, and 5 indicate that the FSL methods successfully adapt to the 6-Way Trench dataset with the best-performing model achieving 96.63% ± 0.26% balanced accuracy in the 20-shot scenario. On the other hand, the methods fail to adapt to the 3-Way Josephson Junction (JJ), and 2-Way Lines & Spaces (L&S) datasets in all scenarios, with the best-performing models achieving 52.41% ± 1.79% and 38.89% ± 1.70% balanced accuracies in the 20-shot scenario respectively. A model randomly assigning labels, where the accuracy it obtains can be described as 100/N-Way, would achieve 16.67% balanced accuracy on Trench, 33.34% balanced accuracy on Josephson Junction, and 50% balanced accuracy on Lines & Spaces. Comparing these to the best-performing trained models on JJ and L&S signifies ineffective learning and motivates an investigation into the potential reasons behind the poor performance. Furthermore, for the 50 evaluations in each scenario, an average confusion matrix (ACM) is computed to display the general behaviors of a model and common errors in a specific dataset. The ACM generated for the best-performing model per dataset in Fig 7.

## 3.3 Discussion

The Trench dataset yields a much stronger model performance than the JJ and L&S datasets. Given that Trench is specifically engineered to test defect detection algorithms, it can mean that Trench does not have the same natural noise and characteristics as present in the JJ and L&S datasets, which are derived from real products. This synthetic nature of the Trench dataset could explain its better performance. Looking at the performance on the other datasets, misclassification patterns observed in JJ and L&S may stem from several factors. First, the pre-processed image size of 84×84 pixels used for

training and evaluation could have limited the granularity of visual information, as downsizing often degrades performance and introduces label noise [17, 18]. Second, the use of a small, randomly initialized backbone like Conv-64F, rather than a larger pretrained model such as ResNet-18, could have further hindered performance. Finally, randomness in tasks when sampling for classes in the meta-learning process may have created poorly balanced tasks, for example, those with only defective or non-defective images, or with images from entirely different products. This could cause the model to focus on product classification rather than defect detection, diverging from the intended objective.

## 4 Conclusion

This article evaluates multiple few-shot learning methods for industrial defect detection on real-world scanning electron microscopy (SEM) images. Through testing, we were able to highlight the best performing methods; specifically *Baseline*, and identify potential problems in using the methods where the models did not perform effectively on the Josephson Junction and the Lines & Spaces dataset but perform comparatively well on the Trench dataset. The underlying random sampling technique and/or characteristic differences in the datasets may explain this variability in performance, although this requires further investigation.

## Acknowledgments

This work was supported by [EMPLOYEE OBSCURED FOR BLIND REVIEW] at [COMPANY OBSCURED FOR BLIND REVIEW], who created the Trench, Josephson Junction, and Lines & Spaces datasets. Furthermore, a significant portion of the research and writing presented in this paper is based on the master's thesis of the [CORRESPONDING AUTHOR].

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

**Table 3.** Few-shot classification accuracies of the Conv-64F backbone on the Trench dataset are reported with mean balanced accuracy and 95% confidence intervals. A random classifier would achieve 16.66%. The best model is shown in bold, and the second-best in red. Meta-learning models are constrained by their training scenario (e.g., a 5-Way model can only be evaluated in 5-Way settings or less). To allow evaluation on the 6-Way Trench dataset, meta-learning models are instead trained in 6-Way scenarios*.

| Model | 6-Way-1-Shot | 6-Way-5-Shot | 6-Way-10-Shot | 6-Way-20-Shot |
|---|---|---|---|---|
| Baseline 5-Way-1-Shot | 47.59% ± 1.11% | 74.03% ± 2.29% | 64.61% ± 4.42% | 59.50% ± 2.90% |
| Baseline 5-Way-5-Shot | **54.97% ± 1.72%** | **85.64% ± 0.87%** | **92.9% ± 0.40%** | **96.63% ± 0.26%** |
| SKD Gen 0 5-Way-1-Shot | 53.25% ± 1.76% | 65.9% ± 1.33% | 72.85% ± 0.76% | 78.02% ± 0.50% |
| SKD Gen 0 5-Way-5-Shot | 48.65% ± 2.41% | 65.07% ± 1.18% | 71.48% ± 0.82% | 77.37% ± 0.57% |
| MAML 6-Way-1-Shot* | 35.95% ± 1.63% | 55.89% ± 2.48% | 60.44% ± 1.74% | 63.30% ± 1.40% |
| MAML 6-Way-5-Shot* | 27.52% ± 1.00% | 49.35% ± 2.47% | 56.26% ± 1.56% | 58.97% ± 1.12% |
| BOIL 6-Way-1-Shot* | 45.32% ± 2.01% | 58.99% ± 1.53% | 62.02% ± 0.85% | 62.66% ± 0.61% |
| BOIL 6-Way-5-Shot* | 47.21% ± 1.82% | 62.32% ± 1.35% | 66.21% ± 0.87% | 69.21% ± 0.80% |
| ProtoNet 5-Way-1-Shot | 49.75% ± 2.77% | 58.71% ± 1.72% | 63.95% ± 1.45% | 67.19% ± 0.75% |
| ProtoNet 5-Way-5-Shot | 39.22% ± 1.58% | 51.21% ± 1.77% | 54.65% ± 1.62% | 56.53% ± 1.35% |
| RENet 5-Way-1-Shot | 47.00% ± 2.48% | 61.36% ± 1.47% | 64.81% ± 1.05% | 68.71% ± 0.75% |
| RENet 5-Way-5-Shot | 41.19% ± 2.68% | 52.70% ± 1.42% | 53.35% ± 0.55% | 54.50% ± 0.36% |

**Table 4.** Few-shot classification accuracies of the Conv-64F backbone on the Lines & Spaces dataset are reported with mean balanced accuracy and 95% confidence intervals. A random classifier would achieve 16.66%. The best model is shown in bold, and the second-best in red. Meta-learning models are constrained by their training scenario (e.g., a 5-Way model can only be evaluated in 5-Way settings or less). To allow evaluation on the 6-Way Trench dataset, meta-learning models are instead trained in 6-Way scenarios*.

| Model | 2-Way-1-Shot | 2-Way-5-Shot | 2-Way-10-Shot | 2-Way-20-Shot |
|---|---|---|---|---|
| Baseline 5-Way-1-Shot | 50.40% ± 5.91% | 50.78% ± 3.08% | 51.27% ± 2.46% | 50.6% ± 1.63% |
| Baseline 5-Way-5-Shot | 50.02% ± 6.94% | **51.81% ± 3.06%** | 51.19% ± 2.17% | **52.41% ± 1.79%** |
| SKD Gen 0 5-Way-1-Shot | 50.07% ± 4.62% | 50.92% ± 2.69% | 50.92% ± 2.83% | 51.33% ± 1.84% |
| SKD Gen 0 5-Way-5-Shot | 50.23% ± 4.79% | 50.07% ± 3.36% | 49.26% ± 2.75% | 49.83% ± 2.37% |
| MAML 6-Way-1-Shot* | 49.69% ± 3.00% | 51.31% ± 5.00% | **51.29% ± 3.08%** | 50.84% ± 2.05% |
| MAML 6-Way-5-Shot* | **51.67% ± 3.53%** | 50.50% ± 3.94% | 51.10% ± 2.55% | 50.74% ± 2.02% |
| BOIL 6-Way-1-Shot* | 49.98% ± 4.03% | 50.86% ± 3.41% | 50.69% ± 2.98% | 50.57% ± 2.83% |
| BOIL 6-Way-5-Shot* | 50.08% ± 3.91% | 50.53% ± 3.01% | 50.90% ± 2.64% | 51.08% ± 2.81% |
| ProtoNet 5-Way-1-Shot | 49.88% ± 6.57% | 49.88% ± 4.70% | 48.95% ± 3.39% | 50.17% ± 3.64% |
| ProtoNet 5-Way-5-Shot | 50.16% ± 6.08% | 49.16% ± 2.75% | 50.82% ± 2.50% | 50.60% ± 2.46% |
| RENet 5-Way-1-Shot | 49.87% ± 6.50% | 49.72% ± 4.38% | 50.10% ± 3.55% | 49.60% ± 3.49% |
| RENet 5-Way-5-Shot | 49.77% ± 5.51% | 50.57% ± 4.20% | 50.36% ± 3.78% | 50.28% ± 3.07% |

[3] D. Tabernik, S. Šela, J. Skvarč, and D. Skočaj. "Segmentation-Based Deep-Learning Approach for Surface-Defect Detection". In: *Journal of Intelligent Manufacturing* 31.3 (Mar. 2019), pp. 759–776. DOI: 10.1007/s10845-019-01476-x. URL: https://doi.org/10.1007/s10845-019-01476-x.

[4] I. Konovalenko, P. Maruschak, J. Brezinová, J. Viňáš, and J. Brezina. "Steel surface defect classification using deep residual neural network". In: *Metals* 10.6 (June 2020), p. 846. ISSN: 2075-4701. DOI: 10.3390/met10060846.

[5] P. Zajec, J. M. Rožanec, S. Theodoropoulos, M. Fontul, E. Koehorst, B. Fortuna, and D. Mladenić. "Few-shot learning for defect detection in manufacturing". In: *International Journal of Production Research* 62 (19 2024),

pp. 6979–6998. ISSN: 1366588X. DOI: 10.1080/00207543.2024.2316279.

[6] W. Li, Z. Wang, X. Yang, C. Dong, P. Tian, T. Qin, J. Huo, Y. Shi, L. Wang, Y. Gao, and J. Luo. "LibFewShot: A Comprehensive Library for Few-Shot Learning". In: *IEEE Transactions on Pattern Analysis and Machine Intelligence* 45 (12 Dec. 2023), pp. 14938–14955. ISSN: 19393539. DOI: 10.1109/TPAMI.2023.3312125.

[7] W.-Y. Chen, Y.-C. Liu, Z. Kira, Y.-C. F. Wang, and J.-B. Huang. "A Closer Look at Few-shot Classification". In: (Apr. 2019). URL: http://arxiv.org/abs/1904.04232.

[8] J. Rajasegaran, S. Khan, M. Hayat, F. S. Khan, and M. Shah. "Self-supervised Knowledge Distillation for Few-shot Learning". In:

**Table 5.** Few-shot classification accuracies of the Conv-64F backbone on the Josephson Junction dataset are reported with mean balanced accuracy and 95% confidence intervals. A random classifier would achieve 16.66%. The best model is shown in bold, and the second-best in red. Meta-learning models are constrained by their training scenario (e.g., a 5-Way model can only be evaluated in 5-Way settings or less). To allow evaluation on the 6-Way Trench dataset, meta-learning models are instead trained in 6-Way scenarios*.

| Model | 3-Way-1-Shot | 3-Way-5-Shot | 3-Way-10-Shot | 3-Way-20-Shot |
|---|---|---|---|---|
| Baseline 5-Way-1-Shot | **34.75% ± 3.23%** | 33.91% ± 1.45% | 33.74% ± 2.64% | 33.33% ± 0.00% |
| Baseline 5-Way-5-Shot | 33.45% ± 4.03% | 35.83% ± 2.44% | 35.87% ± 1.06% | 33.41% ± 0.04% |
| SKD Gen 0 5-Way-1-Shot | 33.72% ± 2.92% | 34.85% ± 1.97% | 35.18% ± 1.49% | 36.77% ± 1.37% |
| SKD Gen 0 5-Way-5-Shot | 33.56% ± 3.43% | 34.90% ± 2.40% | 36.40% ± 1.77% | 37.59% ± 1.71% |
| MAML 6-Way-1-Shot* | 33.48% ± 5.54% | 33.31% ± 7.80% | 33.31% ± 7.35% | 33.37% ± 8.05% |
| MAML 6-Way-5-Shot* | 33.53% ± 3.59% | 33.77% ± 5.86% | 34.06% ± 6.26% | 34.00% ± 5.93% |
| BOIL 6-Way-1-Shot* | 34.14% ± 2.34% | 35.13% ± 2.12% | 36.97% ± 2.61% | 37.02% ± 2.25% |
| BOIL 6-Way-5-Shot* | 34.60% ± 2.40% | 35.18% ± 2.32% | 36.10% ± 2.59% | 37.93% ± 2.50% |
| ProtoNet 5-Way-1-Shot | 34.72% ± 3.97% | 36.80% ± 1.75% | 36.59% ± 1.59% | 38.55% ± 1.32% |
| ProtoNet 5-Way-5-Shot | 33.89% ± 2.79% | **36.85% ± 1.98%** | **37.42% ± 1.81%** | **38.89% ± 1.70%** |
| RENet 5-Way-1-Shot | 34.25% ± 2.97% | 34.60% ± 2.56% | 34.41% ± 2.35% | 36.61% ± 2.28% |
| RENet 5-Way-5-Shot | 34.19% ± 2.53% | 35.25% ± 1.96% | 35.22% ± 1.65% | 36.46% ± 1.75% |

(June 2020). URL: http://arxiv.org/abs/2006.09785.

[9] C. Finn, P. Abbeel, and S. Levine. "Model-Agnostic Meta-Learning for Fast Adaptation of Deep Networks". In: (Mar. 2017). URL: http://arxiv.org/abs/1703.03400.

[10] J. Oh, H. Yoo, C. Kim, and S.-Y. Yun. "BOIL: Towards Representation Change for Few-shot Learning". In: (Aug. 2020). URL: http://arxiv.org/abs/2008.08882.

[11] J. Snell, K. Swersky, and T. R. Zemel. "Prototypical Networks for Few-shot Learning". In: (2017).

[12] A. Paszke, S. Gross, F. Massa, A. Lerer, J. Bradbury, G. Chanan, T. Killeen, Z. Lin, N. Gimelshein, L. Antiga, A. Desmaison, A. Köpf, E. Yang, Z. DeVito, M. Raison, A. Tejani, S. Chilamkurthy, B. Steiner, L. Fang, J. Bai, and S. Chintala. "PyTorch: An Imperative Style, High-Performance Deep Learning Library". In: (Dec. 2019). URL: http://arxiv.org/abs/1912.01703.

[13] D. Kang, H. Kwon, J. Min, and M. Cho. "Relational Embedding for Few-Shot Classification". In: (Aug. 2021). URL: http://arxiv.org/abs/2108.09666.

[14] Y. Song, T. Wang, P. Cai, S. K. Mondal, and J. P. Sahoo. "A Comprehensive Survey of Few-shot Learning: Evolution, Applications, Challenges, and Opportunities". In: *ACM Computing Surveys* 55 (13s July 2023). ISSN: 15577341. DOI: 10.1145/3582688.

[15] J. S. Akosa. "Predictive Accuracy: A Misleading Performance Measure for Highly Imbalanced Data". In: *Proceedings of the SAS Global Forum*. 2017.

[16] P. Bergmann, M. Fauser, D. Sattlegger, and C. Steger. "MVTec AD-A Comprehensive Real-World Dataset for Unsupervised Anomaly Detection". In: (2019). URL: www.mi.imati.cnr.it/ettore/NanoTWICE/.

[17] M. Balaji, J. J. Luke, and R. Joseph. "Impact of Image Size on Accuracy and Generalization of Convolutional Neural Networks". In: *IJRAR19SP012 International Journal of Research and Analytical Reviews* (2019). ISSN: 2349-5138. URL: www.ijrar.org.

[18] V. Thambawita, I. Strümke, S. A. Hicks, P. Halvorsen, S. Parasa, and M. A. Riegler. "Impact of image resolution on deep learning performance in endoscopy image classification: An experimental study using a large dataset of endoscopic images". In: *Diagnostics* 11 (12 Dec. 2021). ISSN: 20754418. DOI: 10.3390/diagnostics11122183.

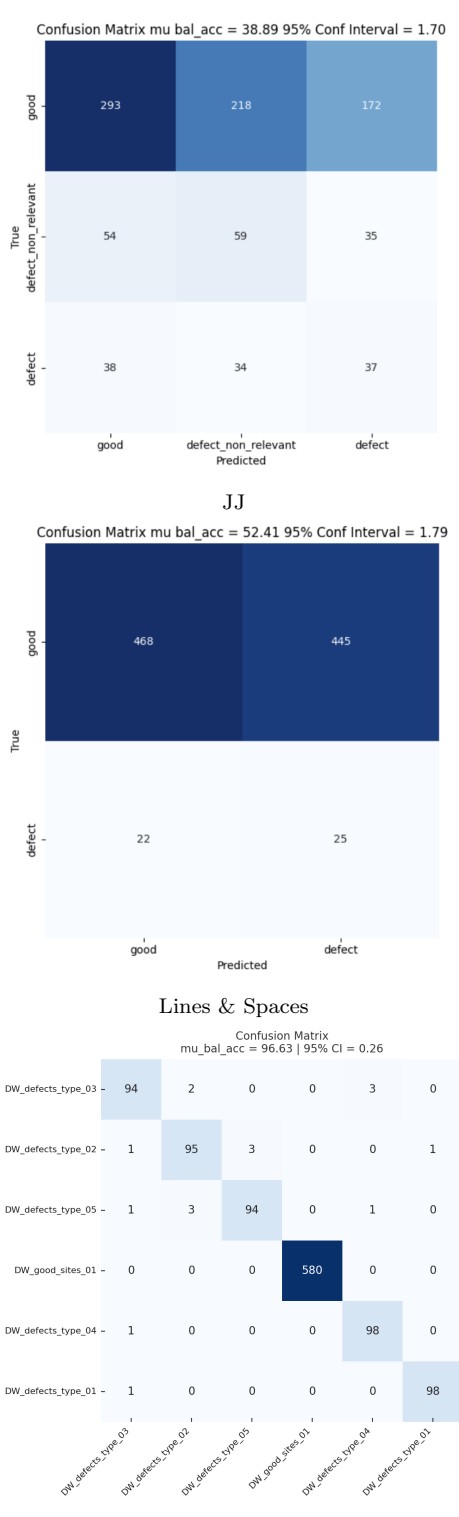

**Figure 7.** Average confusion matrices of the best-performing model over 50 evaluation loops for each evaluation dataset.

