# OpenReview forum: "Few-Shot Learning for Industrial Defect Detection on Novel Scanning Electron Microscopy Datasets"
_NLDL.org/2026/Conference — Submitted to NLDL 2026_

### Official Review · Reviewer_7Jcc · 2025-09-22
**Revision 1**

**Rating:** 2
**Confidence:** 4

**Summary:**

This paper evaluates Few-Shot Learning (FSL) methods on a cross-domain Industrial Defect Detection (IDD) task, pre-training on optical images and testing on Scanning Electron Microscope (SEM) data. The authors contribute three new SEM datasets for this purpose and propose an evaluation framework more aligned with industrial deployment. The key finding is that while the tested FSL methods succeed on one dataset with synthetic-style defects, they fail completely on two more realistic datasets, performing at or below random chance. The paper speculates on the reasons for this failure, including model capacity and image resolution.

**Strengths:**

The primary strength is the introduction of three new, public SEM datasets. This is a significant contribution to a data-scarce field and will enable future benchmarking.

The paper addresses a relevant industrial problem (cross-domain IDD) and proposes a sensible evaluation framework that better reflects a real-world deployment scenario.

The authors are commendable for openly reporting and discussing the negative results, which is a valuable scientific practice.

**Weaknesses:**

The paper's core weakness is its failure to investigate why the methods fail. It identifies a problem but provides no diagnostic experiments (e.g., ablations on image size, backbone architecture). This leaves the reader with an observation rather than an insight.

The experimental setup undermines the validity of the conclusions. The use of a tiny, scratch-trained Conv64F backbone is insufficient for a challenging cross-domain task. Downsampling detailed SEM images to 84x84 likely destroys the very features required for defect detection. These choices, rather than the FSL algorithms themselves, are the most probable cause of the observed failure.

Without further experiments, the paper cannot convincingly claim that state-of-the-art FSL is unsuitable for this task; it can only claim that this specific, limited setup failed.


Can you provide results using a standard ResNet backbone to demonstrate that the failure is not simply due to insufficient model capacity?

How does performance change at a more appropriate image resolution (e.g., 224x224)?

Without these experiments, how can you be sure the failure is inherent to the FSL methods and not just your experimental setup?

**Justification:**

The paper offers a new dataset, but it is critically flawed and lacks support for its conclusions. It shows that a constrained experimental setup (tiny backbone, low-resolution images) fails in a cross-domain task, without adequately diagnosing the cause through ablation studies. Consequently, it does not contribute meaningfully to the understanding of FSL, posing a question without providing answers. The claims are unconvincing, as the poor results seem due to the flawed design rather than limitations of FSL methods.

---

> ### Author Rebuttal · Authors · 2025-10-22
>
> We thank the reviewer(s) for these insightful comments. While our primary goal in this work was to provide an initial, controlled exploration of cross-domain transferability from optical to SEM images, we agree that additional diagnostic experiments—such as ablations on backbone size or input resolution—would provide deeper insights.
>
> In the revised manuscript, we will clarify that the presented results are specific to this controlled setting and are intended as an initial exploration. We view this exploratory step as valuable in its own right, serving as a foundation for more comprehensive future investigations rather than a definitive assessment of state-of-the-art FSL performance on SEM imagery.

---

### Official Review · Reviewer_SUfm · 2025-10-06
**Few-Shot Learning for Industrial Defect Detection on Novel Scanning Electron Microscopy Datasets**

**Rating:** 2
**Confidence:** 4

**Summary:**

This paper investigates the performance of six state-of-the-art few-shot learning (FSL) methods for industrial defect detection tasks on scanning electron microscopy (SEM) images. The authors introduce three new SEM datasets and propose an updated evaluation protocol. Models are pre-trained on optical datasets and evaluated on these SEM datasets to examine cross-domain generalization capabilities. Results show strong performance on a synthetic “Trench” dataset but poor generalization on two real-world datasets (“Josephson Junction” and “Lines & Spaces”).

**Strengths:**

This paper tackles a relevant and timely problem in industrial defect detection — the application of FSL techniques to SEM images. Its primary strength lies in addressing the challenge of limited labeled data in industrial settings by benchmarking a range of established FSL methods, including transfer learning, meta-learning, and metric learning approaches.
Therefore, another contribution is the introduction of three SEM datasets representing different defect types and imaging conditions, which could serve as valuable resources for future research in this domain if made publicly available. Furthermore, the authors propose an evaluation framework tailored to industrial tasks, employing balanced accuracy to account for class imbalance — a critical consideration in defect detection — and adjusting the evaluation procedure to better reflect real-world scenarios.

Thus, overall, the paper is clearly written, logically structured, and provides useful information on the limitations and capabilities of current FSL techniques in cross-modality industrial applications.

**Weaknesses:**

Despite these strengths, the paper suffers from significant limitations that reduce its novelty and impact.
The most critical weakness is the lack of a methodological contribution: the work only benchmarks existing FSL algorithms without proposing new techniques, improvements, or domain adaptation strategies. Given that recent works like SEM-CLIP already explore few-shot defect detection in SEM imagery, this significantly limits the paper’s originality.
Moreover, the results are underwhelming — models fail to achieve meaningful performance on two of the three real-world SEM datasets, barely surpassing random baselines. While the authors acknowledge possible causes such as low image resolution and shallow network architectures, they do not attempt further experiments (e.g., larger backbones, higher resolutions) to validate these claims.
Thus, the paper also lacks comparison with more recent, SEM-specific approaches, leaving its benchmarking context incomplete.

Finally, although the paper repeatedly mentions the release of three new SEM datasets, it does not provide any download links or repository information, making it unclear whether these datasets will actually be accessible to the research community, which severely limits the value of one of its main claimed contributions.

**Justification:**

While the paper addresses a relevant and important problem — few-shot defect detection in SEM images — it lacks sufficient novelty and technical depth to warrant publication in its current form. The work primarily benchmarks existing few-shot learning algorithms without proposing any new methodology, framework, or adaptation strategy, which significantly limits its contribution to the field.

Moreover, recent research, such as SEM-CLIP, has already explored this problem with more innovative approaches, raising concerns about the originality of this submission. The results presented are also underwhelming, with most models failing to achieve meaningful performance on real-world SEM datasets, and the paper does not conduct deeper experiments to investigate or mitigate these shortcomings.

Additionally, although the introduction of three new SEM datasets is claimed as a key contribution, the paper does not provide any publicly accessible dataset links or clear release plans, which severely diminishes the value and reproducibility of the work. Taken together, the lack of methodological innovation, limited experimental depth, unclear dataset availability, and overlap with prior work justify rejection at this time.

---

> ### Author Rebuttal · Authors · 2025-10-22
>
> Thank you for taking the time to thoroughly review our paper and providing detailed feedback. Here is our rebuttal to the points raised in the review process:
> 1. Regarding the dataset planned to be released, we intentionally did not provide a link to respect the double blind review process. If accepted, the final paper will contain a link to a GitHub repository with the dataset description and a download link for interested parties.
>
> 2. We agree with the reviewer that our paper makes no strong methodological contributions. Our objective in this work is not to propose a new FSL algorithm, but rather to present an initial, controlled exploration of how existing FSL methods perform in this new and challenging context for industrial applications. Guaranteed, further experiments with larger image sizes and a pre-trained backbone could possibly provide a better perspective. We are of the opinion that our results provide value in setting up the foundation for further investigations in this field.
>
> 3. We are also thankful to the reviewer(s) for bringing our focus onto the SEM-CLIP paper, which we were previously unaware of. It seems to be highly relevant to our work, and we intend to incorporate references to it as minor revisions before the final submission.

---

### Official Review · Reviewer_Gg9Z · 2025-10-09
**Well written application paper for few shot learning in defect detection for SEM images.**

**Rating:** 4
**Confidence:** 3

**Summary:**

The manuscript evaluates multiple few shot learning methods for defect detection on scanning electron microscopy images coming from industrial applications in collaboration with stakeholders.

**Strengths:**

- Well written and easy to read
- Good literature and background section
- Task well motivated
- Stability of results is tested wrt to the support sets
- The authors claim to release three electron-beam lithography datasets with this publication
- 6 methods are evaluated for the task

**Weaknesses:**

The three datasets that are one contribution of this study are said to be published with the study, but no link to the dataset is provided in the paper. As this is one of the three contributions stated in the paper, the link to where it can be downloaded has to be included.

The methodological contributions are limited, this is an application paper.

minor remarks:
- lines 99, 102 103: D_{train} and C_{train} contain unnecessary spaces
- Fig 2 and Fig 3,4,5: be consistent with category naming between barplot and images (e.g. “defect_non_relevant” and “irrelevant defect” - I assume the same?; “good” and “non-defect” the same?)
- Fig. 4: “Defect 05” -> “Defect Type 05”?
- Move all figures above the references

**Justification:**

Even though the methodological contributions of the study are limited, the experiment section is thorough and relevant for real world industrial applications, so is the release of public SEM datasets for defect detection.

---

> ### Author Rebuttal · Authors · 2025-10-22
>
> We thank the reviewer(s) for the feedback on the figures and tables. These points will be incorporated in the next minor revision.
>
> ​Regarding the dataset planned to be released, we intentionally did not provide a link to respect the double blind review process. If accepted, the final paper will contain a link to a GitHub repository with the dataset description and a download link for interested parties.

---

### Official Review · Reviewer_N9yL · 2025-10-10
**While the authors contribute three novel datasets for the purpose of assessing few-shot learning techniques for industrial defect detection, they provide little related work as to contrast the proposed evaluation framework and insights against previously proposed ones.**

**Rating:** 2
**Confidence:** 4

**Summary:**

The authors assess multiple few-shot learning techniques across three publicly released datasets to understand their generalization capabilities for industrial defect detection. We consider the research to be sound and mostly correctly framed. Our main observations are that the comparison of few-shot learning techniques has been conducted on three novel datasets, without considering datasets used in few-shot learning research for industrial defect detection, and that the metrics used to assess the quality of the proposed models are not well-defined.

**Strengths:**

We consider the authors' contribution to be valuable, as they have released three novel datasets for industrial defect detection and compared the performance of multiple few-shot learning models across them.

**Weaknesses:**

GENERAL COMMENTS

(1) - The authors compare a wide range of few-shot learning techniques across three datasets. We encourage the authors to include at least one additional dataset from the ones that have been extensively used for few-shot learning assessment, to provide perspective and make their own research comparable to results already published with them. E.g., the authors mention the MVTEC-AD and KolektorSDD datasets.

(2) - The authors provide no related work section. We encourage the authors to introduce such a section detailing (a) few-shot learning techniques, especially those used for industrial defect detection, (ii) benchmark datasets used for industrial defect detection, and (iii) setups used across experiments, so that they can ground their own framework based on previous research works and showcase how their approach is different from the previously used ones.

(3) - Metrics: The authors measure the quality of the models using Accuracy. While they use the balanced version to address class imbalance, the metric itself requires a prior threshold to define True Positives and False Positives. We encourage the authors to consider measuring the model's performance using the AUC-ROC to provide an overall model quality assessment that is threshold-independent.



FIGURES

(4) - Figures 3, 4, 5, 6: We encourage the authors to encircle (e.g., with a red circle) the defects in the images, to help the reader understand what is considered a defect and where the defects are located.

(5) - Figures 1-5 take up a whole page. We encourage the authors to seek alternative representations of class distribution (Figure 2) to convey the same information more compactly.

(6) - Figure 7: We encourage the authors to redraw the diagrams, providing proper names for variables and metrics (avoid using underscores and pruned words, ensuring abbreviations are clarified if necessary).


TABLES

(7) - All tables: align numbers to the right to make differences in magnitude evident.

(8) - Table 3, 4, 5: The tables have some results bolded and some results highlighted in red. We would appreciate some clarification on what those mean for a better interpretation of the results.

**Justification:**

The authors propose three novel datasets for industrial defect identification along with the assessment of multiple few-shot learning techniques, establishing a modest benchmark for this purpose and domain. Nevertheless, the manuscript lacks a proper related-work section that would allow the contributions and novelty of this manuscript to be grounded with respect to previously documented research. Furthermore, the lack of related work makes it difficult to ground some claims about the novelty of the proposed evaluation framework in the context of industrial defect detection.

---

> ### Author Rebuttal · Authors · 2025-10-22
>
> Thank you for taking the time to thoroughly review our work and provide us with valuable feedback. Here is our rebuttal to the points raised in the review process:
>
> 1. Add another common benchmark dataset to provide perspective:
> We agree with reviewers' comments and find them valuable. As one of our main objectives is to study the transferability of representations from publicly available optical images to SEM images, benchmarks based solely on optical imagery do not provide an equivalent or meaningful comparison for our task. Additionally, the Industrial 5i dataset used to train and validate our models also includes MVTEC-AD, KolektorSDD, and other common benchmarks, so this would not be trivial, as we would have to re-run our experiments from scratch.
>
>
> 2. Related work section: We agree that it is valuable to have a dedicated related work section. Though based on previous papers we read from the earlier editions of the conference and the page limits, we found it appropriate to integrate our related work section into section 2 (Methods) and section 3 (Experiments).
>
>
> 3. We appreciate the reviewer’s point regarding the choice of metrics. While alternative metrics could indeed provide clearer model-to-model comparisons, our conclusions are not dependent on such precise comparisons. Rather, they focus on the broader trends derived from the experiments.
>
>
> 4. We thank the reviewer for the feedback on the figures and tables. These points will be incorporated in the next minor revision.

---

### Meta-Review · Area_Chair_tJn1 · 2025-10-31

**Recommendation:** Reject
**Confidence:** 4

**Metareview:**

This paper presents a systematic evaluation of several few-shot learning (FSL) methods for industrial defect detection on novel scanning electron microscopy (SEM) datasets. There is consensus around the presentation of the paper being of a good standard. Probably the main contribution is the three new datasets that could be valuable for future research. It is understandable that several reviewers commented on the fact that there was no link to these datasets. While the authors response is helpful (making those available upon acceptance), there are ways to annonymize repositories that could have been explored.

Reviewers appreciated the relevance of the topic and the openness in reporting negative results and limitations. However, the submission lacks innovation and sometimes fails to fully investigate why the methods fail. The absence of experiments (or hypothesis) to explain this, further weakens the contribution. The authors’ rebuttal clarified the scope as an exploratory benchmark and addressed some presentation concerns, but did not provide additional evidence in this respect which limit the impact of the work in its current form.

---

### Decision · Program_Chairs · 2025-11-05

**Decision:**

Reject

**Comment:**

Based on the reviewers and AC comments, the paper cannot be presented at the conference.